# INQUIRE: INteractive Querying for User-aware Informative REasoning

**Tesca Fitzgerald**[*,1,2]  **Pallavi Koppol**[2]  **Patrick Callaghan**[2]  **Russell Q. Wong**[2]
**Reid Simmons**[2]  **Oliver Kroemer**[2]  **Henny Admoni**[2]
[1] Yale University  [2] Carnegie Mellon University
tesca.fitzgerald@yale.edu,
{pkoppol, pcallagh, rqwong, rsimmons, okroemer, hadmoni}@andrew.cmu.edu

**Abstract:** Research on Interactive Robot Learning has yielded several modalities for querying a human for training data, including demonstrations, preferences, and corrections. While prior work in this space has focused on optimizing the robot's queries within each interaction type, there has been little work on optimizing over the selection of the interaction type itself. We present INQUIRE, the first algorithm to implement and optimize over a generalized representation of information gain across multiple interaction types. Our evaluations show that INQUIRE can dynamically optimize its interaction type (and respective optimal query) based on its current learning status and the robot's state in the world, resulting in more robust performance across tasks in comparison to state-of-the-art baseline methods. Additionally, INQUIRE allows for customizable cost metrics to bias its selection of interaction types, enabling this algorithm to be tailored to a robot's particular deployment domain and formulate cost-aware, informative queries.

**Keywords:** Active Learning, Learning from Demonstration, Human-Robot Interaction

## 1 Introduction

As we envision robots that adapt to novel tasks and environments after deployment, it is important to consider *how* they can efficiently obtain training data to address this novelty. Research in Interactive Robot Learning has yielded many effective methods for obtaining this training data via interaction between a robot and a human teacher. In a *demonstration*, the teacher provides the trajectory that the robot should take starting from a particular state [1, 2]. In a *preference* query, the teacher selects one trajectory from a set of candidates proposed by the robot [3, 4]. In response to a single trajectory proposed by the robot, the teacher can provide a *correction* [5, 6] or simply a *binary reward* [7].

These interaction types differ according to how the robot queries the teacher, how the teacher is constrained in providing feedback, how the robot should interpret the teacher's feedback as training data, and the physical and cognitive load imposed on the teacher [8]. Prior work in Active Learning has investigated how to formulate informative queries by maximizing the expected information gain resulting from the teacher's feedback. However, barring a few exceptions ([9, 10, 11]), prior work typically assumes that the robot uses a single interaction type for all queries. We expect that the *optimal* interaction type depends on the robot's task knowledge (which changes over time), the robot's query state (i.e., the state from which it queries the teacher), and domain-specific considerations (e.g., the time or effort it takes a teacher to respond to queries) [12].

Our work is motivated by this question: How can a robot optimize both the *type* and *content* of its queries to a human teacher based on the information it needs at any given moment? We introduce INQUIRE: a robot learning system that performs this optimization by representing multiple interaction types in a single unified framework, enabling the robot to directly estimate and compare the expected information gain of its queries across multiple interaction types. We evaluated INQUIRE against two state-of-the-art interactive learning methods that use a single or fixed pattern of interaction types. We analyzed the effect of domain on INQUIRE's performance and selection

6th Conference on Robot Learning (CoRL 2022), Auckland, New Zealand.

of interaction types over time by simulating four domains with unique reward-learning problems. We found that INQUIRE learned reward functions that were more accurate and resulted in better task performance than either baseline, with particular strength in accommodating low-information query states (i.e., repeated states in which the robot has already received feedback). Furthermore, we demonstrate how INQUIRE can incorporate cost metrics (representing physical or cognitive load on the teacher), optimizing queries over both the informativeness and ease of the teacher's responses.

## 2 Related Works

Learning from Demonstration involves interpreting human feedback as training data using methods such as imitation learning [1] or inverse reinforcement learning [2]. However, there are many other forms that human feedback can take, including preferences [3, 4], labels [13], and corrections [6, 5]. These approaches optimize queries *within* a single interaction type, typically by maximizing volume removal [14] or information gain resulting from the teacher's response to the query [3]. Other approaches such as min-max regret optimization have also been explored [15]. Prior work has also investigated the use of fixed strategies for selecting interaction types; for example, requesting a fixed number of demonstrations before requesting preferences for the remaining queries [9, 16]. [11] incorporates more interaction types (demonstrations, labels, and feature queries) and contributes both rule-based and decision-theoretic strategies for query selection. Other methods for learning from multiple feedback types include combining preferences with ordinal labels [17], and using both demonstrations and rankings [18]. Furthermore, [19] presents a software library for combining different preference feedback types and demonstrations.

Several attempts have been made to impose a unifying and consistent framework across multiple interaction types. [20] describes interactions in terms of the explicit and implicit information they convey; [8] surveys how different interaction types affect a teacher's ability to provide informative feedback. Our work similarly contributes a perspective on the unifying and differentiating features of interaction types: we propose a generalized framework for computing information gain across multiple interaction types. We focus on four interaction types corresponding to the archetypes (*Showing, Categorizing, Sorting*, and *Evaluating*) identified in [12], and empirically show the effects of dynamically selecting interaction types in robot learning.

## 3 Approach

We define a **query** as a set of possible choices presented to the teacher, and **feedback** as the teacher's selected choice in response to a query. Our goal is to enable a robot to (1) efficiently query a teacher using multiple interaction types, and (2) learn from feedback obtained via these interactions. We ground this goal in the problem of learning a distribution $\mathcal{W}$ over feature weight vectors $\omega \in \mathcal{W}$, each resulting in a linear reward function $r(t) = \phi(t) \cdot \omega$, where $\phi(t)$ is the feature vector of a trajectory $t$. Thus, our goal translates into (1) selecting queries and interaction types that minimize uncertainty over $\mathcal{W}$, and (2) updating $\mathcal{W}$ over feedback from multiple interaction types.

We present INQUIRE (Alg. 1), an algorithm comprised of three key steps for each query: (1) selecting the optimal interaction type $i$ and corresponding query $q_i^*$ that maximizes the information gain over the weight distribution $\mathcal{W}$ (approximated as the sample set $\Omega$), (2) recording the teacher's response to that query (i.e., feedback) in a feedback set $\mathbf{F}$, and (3) updating the weight distribution $\mathcal{W}$ such that it maximizes the likelihood of all feedback in $\mathbf{F}$. To generalize across multiple interaction types, we must contend with the differing formulations of *query* and *feedback* corresponding to each type. We follow the framing presented in [8], where each interaction type consists of a *query space* $Q(s)$ (the set of possible queries from state $s$) and a *choice space* $C(q)$ (the set of possible teacher feedback, i.e., the choices available to the teacher in response to a query $q \in Q(s)$). We assume the robot must query from whatever initial state $s$ it is placed in, and cannot optimize the state $s$ itself.

For a **demonstration**, let $\mathcal{T}(s)$ represent the set of all possible trajectories originating from the initial state $s$. The robot (implicitly) enables the teacher to demonstrate any trajectory in this set, and thus its query space is $Q(s) = \{\mathcal{T}(s)\}$ (i.e., a single query consisting of the entire trajectory space). The teacher's choice space is $C = \mathcal{T}(s)$ (any trajectory within that space). For a **preference**, the robot queries the teacher with two trajectories $q = \{t_0, t_1 \mid t_0, t_1 \in \mathcal{T}(s)\}$ who then chooses either $t_0$ or $t_1$. The query space is $Q(s) = \mathcal{T}(s) \text{ x } \mathcal{T}(s)$ and the teacher's choice space is $C(q) = \{t_0, t_1\}$.

Table 1: Each interaction involves separate query spaces, choice spaces, and choice implications.

| | Query Space $Q_i(s)$ | Query $q \in Q_i(s)$ | Choice Space $C_i(q)$ | Choice Implication $c \in C_i(q) \implies (c^+, c^-)$ |
|---|---|---|---|---|
| **Demo.** | $\{T\}$ | $T$ | $T$ | $c^+ : t \in T \quad c^- : T \setminus t$ |
| **Pref.** | $T \times T$ | $\{t_0, t_1\}, t_0, t_1 \in T$ | $\{t_0, t_1\}$ | $c^+ : t \in q \quad c^- : q \setminus c^+$ |
| **Corr.** | $T$ | $t \in T$ | $T$ | $c^+ : t' \in T \quad c^- : q$ |
| **Binary** | $T$ | $t \in T$ | $\{0, 1\}$ | $c = 0 \implies c^+ : T \setminus q \quad c^- : q$ 
 $c = 1 \implies c^+ : q \quad c^- : T \setminus q$ |

---

**Algorithm 1** INQUIRE - Overview

**Input**: Set of query states $S$
**Parameters**: $K$ (# of queries), $\mathcal{I}$ (interaction types)
**Output**: Weight vector $\omega^*$

1: $\mathbf{F} \leftarrow \{\}$
2: $\mathbf{\Omega} \leftarrow M$ random initial weight vectors
3: **for** $K$ iterations **do**
4: $\quad s \leftarrow$ next query state in $S$
5: $\quad q_i^* \leftarrow$ generate_query$(s, \mathcal{I}, \mathbf{\Omega})$ **(Alg. 2)**

6: $\quad \mathbf{F} \leftarrow \mathbf{F} \cup \{$query_teacher$(q_i^*)\}$
7: $\quad \mathbf{\Omega} \leftarrow$ update_weights$(\mathbf{F})$
8: $\omega^* \leftarrow$ mean$(\mathbf{\Omega})$
9: **return** $\omega^*$

---

**Algorithm 2** INQUIRE - Generate Query

**Input**: $s$ (state), $\mathcal{I}$ (interaction types), $\mathbf{\Omega}$ (weight samples)
**Output**: Query $q^*$

1: $\mathbf{T} \leftarrow$ uniformly_sample_trajectories$(s)$
2: Compute $\mathbf{E} : \{\mathbf{E}_{t,t',\omega}, \forall t, t' \in \mathbf{T}, \omega \in \mathbf{\Omega}\}$ **(Eq. 4)**
3: **for** each interaction type $i \in \mathcal{I}$ **do**
4: $\quad \mathbf{Q} \leftarrow Q_i(s)$ **(See Table 1)**
5: $\quad \mathbf{C} \leftarrow \{C_i(q), \forall q \in \mathbf{Q}\}$ **(See Table 1)**
6: $\quad$ Compute info gain matrix $\mathbf{G}^{(\mathbf{i})}$ from $\mathbf{E}$ **(Eq. 9)**
7: $\quad q \leftarrow \arg\max_{q'} \sum_{c \in \mathbf{C}_{q'}, \omega \in \mathbf{\Omega}} \mathbf{G}^{(i)}_{q',c,\omega}$
8: $\quad g \leftarrow \frac{1}{\log(\lambda_i)} \sum_{c \in \mathbf{C}_q, \omega \in \mathbf{\Omega}} \mathbf{G}^{(i)}_{q,c,\omega}$
9: $\quad$ **if** information gain $g > g^*$ **then**
10: $\quad\quad g^* \leftarrow g$
11: $\quad\quad q^* \leftarrow q$ {Store query with highest info. gain}
12: **return** $q^*$

---

For a **correction**, the robot executes one trajectory $q \in \mathcal{T}(s)$ which the teacher then modifies to a preferable behavior. The agent's query space is $Q(s) = \mathcal{T}(s)$ and the teacher's choice space is $C(q) = \mathcal{T}(s)$. For **binary reward**, the robot executes a single trajectory $q \in \mathcal{T}(s)$, and the teacher indicates a positive or negative reward. The agent's query space is $Q(s) = \mathcal{T}(s)$ and the teacher's choice space is $C(q) = \{0, 1\}$.

The *implication* of the teacher's choice $c \in C(q)$ is a set of accepted trajectories $c^+$ and set of rejected trajectories $c^-$, which we define in Table 1 and use later to calculate information gain. Since the set of all possible trajectories originating from $s$ (represented by $\mathcal{T}(s)$) is potentially infinite, we approximate it as the set $T$ containing $N$ trajectory samples originating from the state $s$ and consisting of randomly selected actions.

## 3.1 Query Optimization

When optimizing the agent's query, our goal is to greedily select one that maximizes the agent's expected information gain over $\mathcal{W}$ after receiving any feedback from the choice set (summarized in Alg. 2). Selecting a query involves optimizing over information gain (IG) as follows:

$$q_i^* = \arg\max_{q \in Q_i(s)} \mathbb{E}_{c|C_i(q)} [\text{IG}(\mathcal{W} \mid c)] \tag{1}$$

$$= \arg\max_{q \in Q_i(s)} \sum_{c \in C_i(q)} \sum_{w \in \Omega} \left[ P(c|w) \cdot \log \frac{M \cdot P(c|w)}{\sum_{w' \in \Omega} P(c|w')} \right] \tag{2}$$

where $\Omega$ contains $M$ samples of the distribution $\mathcal{W}$. The expansion from Eq. 1 to 2 follows the derivation presented in [3]; see Appendix A.1 for intermediate steps. We adopt the commonly-used Boltzmann-rational equation to define $P(c|\omega)$:

$$P(c|\omega) = \frac{\sum_{t \in c^+} e^{\beta \cdot \phi(t) \cdot \omega}}{\sum_{t \in c^+ \cup c^-} e^{\beta \cdot \phi(t) \cdot \omega}} \tag{3}$$

where $\phi(t)$ returns the feature trace of the trajectory $t$; that is, the sum over the feature vectors of all states visited in trajectory $t$.[1] Note that Eq. 3 reduces to Bayesian Inverse Reinforcement Learning [21] for each $t \in c^+$. $\beta$ is a parameter representing the expected optimality of the teacher's feedback with respect to $\omega$. We assign a value of $\beta = 20$ across all interaction types (selected through empirical evaluation).

To minimize the computational complexity of solving for Eq. 2, we reformulate it as a series of operations over a $|Q|$ x $|C|$ x $|\Omega|$ probability tensor $\mathbf{P}$, where $\mathbf{P}_{q,c,\omega}$ represents the probability (according to weight sample $\omega \in \Omega$) that the teacher will select choice $c$ in response to query $q$. To construct $\mathbf{P}$, let $\mathbf{E}$ be a $N$ x $N$ x $M$ (i.e., $|T|$ x $|T|$ x $|\Omega|$) tensor representing exponentiated rewards:

$$\mathbf{E}_{t,t',\omega} = e^{\beta \cdot \phi(t') \cdot \omega} \qquad \implies \qquad \left[\mathbf{E} + \mathbf{E}^{\mathbf{T}}\right]_{t,t',\omega} = e^{\beta \cdot \phi(t') \cdot \omega} + e^{\beta \cdot \phi(t) \cdot \omega} \qquad (4)$$

All tensor transposes are performed over the first two axes. With $\mathbf{E}$ in hand, we next define the probability tensors of each interaction type as follows:

$$\mathbf{P}_{q,c,\omega}^{(\text{demo})} = \left[\mathbf{E}_0 \oslash \sum_{t \in T} \mathbf{E}^{\mathbf{T}}{}_t\right]_{c,\omega} \qquad \text{(since } |Q| = 1 \text{ for demonstrations)} \qquad (5)$$

$$\mathbf{P}_{q,c,\omega}^{(\text{pref})} = \left[\left(\mathbf{E} \oslash (\mathbf{E} + \mathbf{E}^{\mathbf{T}})\right)^{\mathbf{T}}, \mathbf{E} \oslash (\mathbf{E} + \mathbf{E}^{\mathbf{T}})\right]_{c,q_0,q_1,\omega} \qquad (c \in \{0, 1\} \text{ for prefs.)} \qquad (6)$$

$$\mathbf{P}_{q,c,\omega}^{(\text{corr})} = \left[\mathbf{E} \oslash (\mathbf{E} + \mathbf{E}^{\mathbf{T}})\right]_{q,c,\omega} \qquad (7)$$

$$\mathbf{P}_{q,c,\omega}^{(\text{bnry})} = \left[1 - \left(\mathbf{E}_0 \oslash \alpha \sum_{t \in T} \mathbf{E}^{\mathbf{T}}{}_t\right), \mathbf{E}_0 \oslash \alpha \sum_{t \in T} \mathbf{E}^{\mathbf{T}}{}_t\right]_{c,q,\omega} \qquad (c \in \{0, 1\} \text{ for binary rewards)} \quad (8)$$

where $\oslash$ represents an element-wise division of two matrices (i.e., $(\mathbf{A} \oslash \mathbf{B})_{ij} = \mathbf{A}_{ij}/\mathbf{B}_{ij}$) and $\alpha$ is a normalization factor such that $\sum_c \mathbf{P}_{q,c,\omega}^{(\text{bnry})} = 1$. For derivations, see Appendix A.3. **The main effect of this formulation is that it enables tractable optimization over multiple interaction types** by sharing a common representation $\mathbf{E}$. To solve for the optimal query $q_i^*$ using interaction type $i$, we use $\mathbf{P}^{(i)}$ to construct a $|Q|$ x $|C|$ x $|\Omega|$ information gain tensor $\mathbf{G}^{(i)}$:

$$\mathbf{G}_{q,c,\omega}^{(i)} = \mathbf{P}_{q,c,\omega}^{(i)} \cdot \log\left(\frac{M \cdot \mathbf{P}_{q,c,\omega}^{(i)}}{\sum_{\omega' \in \Omega} \mathbf{P}_{q,c,\omega'}^{(i)}}\right) \qquad\qquad q_i^* = \arg\max_q \sum_{c,\omega} \mathbf{G}_{q,c,\omega}^{(i)} \qquad (9)$$

We then solve for the optimal interaction type itself. To perform a *cost-weighted* optimization, with the aim of optimizing over both interaction cost and informativeness, $\lambda_i$ may be set according to domain-specific cost factors (e.g., the time or mental load involved in answering a query) for each interaction type.[2] To perform an *unweighted* optimization and maximize solely over the informativeness of each query, let $\lambda_i$ be a constant value over all interaction types $i \in \mathcal{I}$.

$$i^* = \arg\max_{i \in \mathcal{I}} \frac{1}{\log(\lambda_i)} \sum_{c,\omega} \mathbf{G}_{q_i^*,c,\omega}^{(i)} \qquad (10)$$

We summarize this process in Alg. 2.

### 3.2 Update Weights from Feedback

After presenting the optimal query to the teacher, the agent receives feedback and appends it to a feedback set $\mathbf{F}$—a cumulative set that contains all feedback received by the agent thus far. Our goal is to then update the weight estimate such that it maximizes the likelihood of all feedback in $\mathbf{F}$:

$$\omega^* = \arg\max_\omega \prod_{c \in \mathbf{F}} P(c|\omega) = \arg\max_\omega \prod_{c \in \mathbf{F}} \frac{\sum_{t \in c^+} e^{\beta \cdot \phi(t) \cdot \omega}}{\sum_{t \in c^+ \cup c^-} e^{\beta \cdot \phi(t) \cdot \omega}} \qquad (11)$$

---

[1]See Appendix B.1 for each domain's definition of $\phi$.

[2]In our evaluations, we assign a cost of 20 to each demonstration, 15 to each correction, 10 to each preference, and 5 to each binary query.

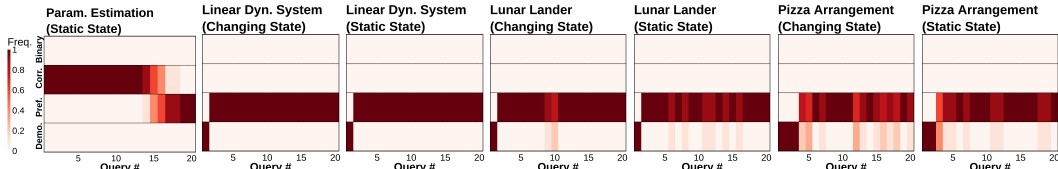

(a) Selected interaction types *without* cost-weighting

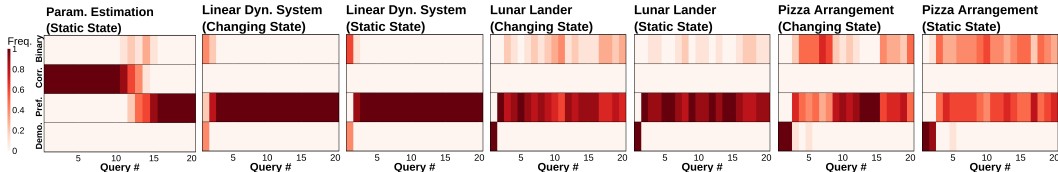

(b) Selected interaction types *with* cost-weighting

Figure 1: Heatmaps illustrating how INQUIRE selects different interaction types as it learns more over time. These selections differ when deriving unweighted (top) or cost-weighted (bottom) information gain estimations. In the cost-weighted setting (bottom), INQUIRE selects more low-cost binary queries than it does in the unweighted setting (top).

We calculate the gradient over $\omega$ by differentiating over its log-likelihood given $\mathbf{F}$:

$$\frac{\partial \ell(\omega)}{\partial \omega_j} = \sum_{c \in \mathbf{F}} \left[ \frac{\sum_{t \in c^+} \beta \cdot \phi_j(t) \cdot e^{\beta \cdot \phi(t) \cdot \omega}}{\sum_{t \in c^+} e^{\beta \cdot \phi(t) \cdot \omega}} - \frac{\sum_{t \in c^+ \cup c^-} \beta \cdot \phi_j(t) \cdot e^{\beta \cdot \phi(t) \cdot \omega}}{\sum_{t \in c^+ \cup c^-} e^{\beta \cdot \phi(t) \cdot \omega}} \right] \tag{12}$$

$$= \sum_{c \in \mathbf{F}} \left[ \beta \cdot \phi_j(c_0^+) - \frac{\sum_{t \in c^+ \cup c^-} \beta \cdot \phi_j(t) \cdot e^{\beta \cdot \phi(t) \cdot \omega}}{\sum_{t \in c^+ \cup c^-} e^{\beta \cdot \phi(t) \cdot \omega}} \right] \qquad (\text{iff } |c^+| = 1) \tag{13}$$

See Appendix A.4 for the full derivation. After receiving feedback from each query and updating $\mathbf{F}$, we approximate $\Omega$ by randomly initializing and then performing gradient ascent on each weight sample $\omega \in \Omega$.

## 4 Results

We simulate four types of learning problems in robotics using an oracle teacher to obtain controlled evaluations. The oracle teacher, similar to INQUIRE, requires its own set of trajectory samples $T'$. It then selects a response to a query via one of three mechanisms: returning the highest-reward trajectory from its choice space (demonstrations/preferences), rejection sampling of trajectories followed by selection of the trajectory with the highest reward-to-distance ratio from the queried trajectory (corrections), and returning whether a query meets or exceeds a reward threshold (binary reward). Implementation details can be found in Appendix B.2.

The **Parameter Estimation** domain involves directly estimating a randomly-initialized, ground truth weight vector $\omega^*$ containing 8 parameters. The **Linear Dynamical System** domain, inspired by [3], simulates a controls problem and involves learning 8 parameters. The **Lunar Lander** domain [22] simulates a controls problem involving 4 parameters. The **Pizza Arrangement** domain simulates a preference-learning problem involving 4 parameters. Each domain (except for Parameter Estimation) has a *static*-state and *changing*-state condition indicating whether the robot must formulate all queries from the same query state or not, respectively. For the full evaluation procedure and oracle implementation details for each domain see Appendix B.

### 4.1 INQUIRE Query Selection

We first analyze how INQUIRE selects queries. Figure 1 reflects the changes in interaction types selected by INQUIRE over time. Figure 1a first reports these interaction selections in an unweighted query optimization setting, where all interaction types are assumed to be equally costly. In the parameter optimization domain, INQUIRE requests corrections in the first 14-18 queries and then requests preferences as the remaining queries. Demonstrations were not enabled in this domain. In

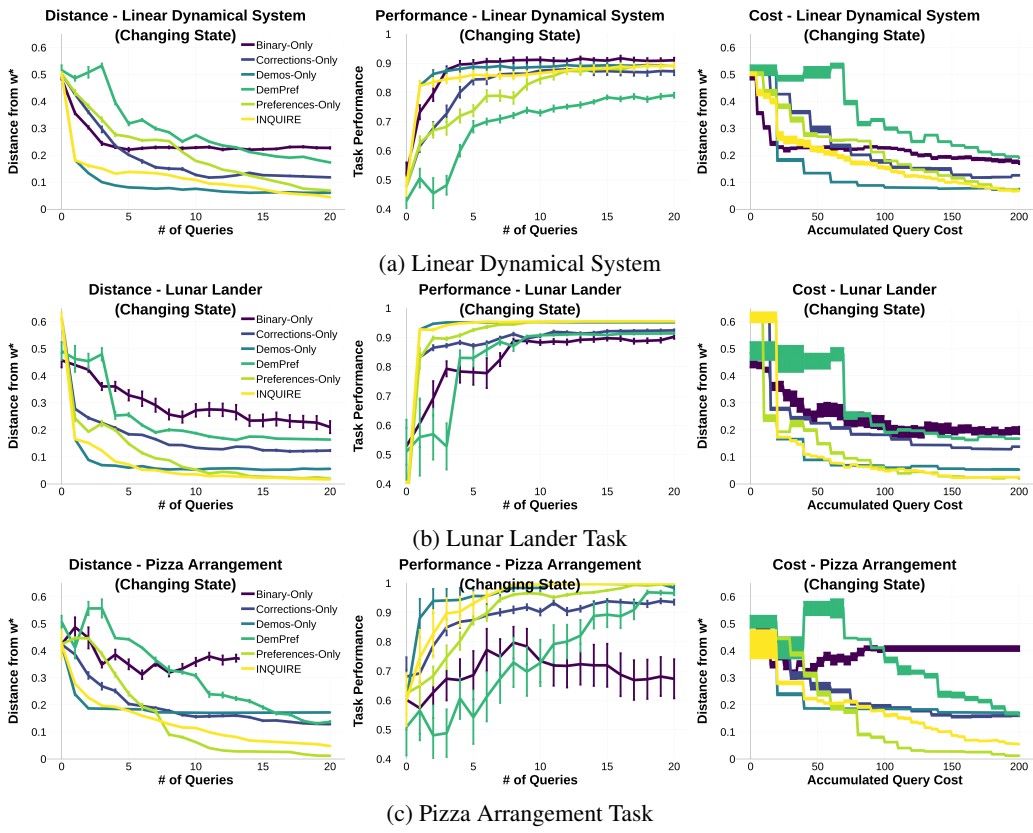

(a) Linear Dynamical System

(b) Lunar Lander Task

(c) Pizza Arrangement Task

Figure 2: Metrics for the *changing state* condition in which the robot's initial state changes with each query. Error bars/regions represent variance across multiple evaluation runs with randomized query states and initial weights. Cost metrics are cut off after 20 queries for the *binary-only* method in (c) due to extensive computation times.

all other domains, INQUIRE requests a demonstration as its first query, then immediately switches to requesting preferences for the remaining queries (occasionally alternating between preferences and demonstrations in the Lunar Lander domain).

After assigning different cost values to each interaction type, INQUIRE chooses more diverse interaction types in order to maximize its information-to-cost ratio. As shown in Figure 1b, this typically results in INQUIRE posing more binary queries due to their relatively low cost. This pivot toward binary queries may occur at the start (as seen in the linear dynamical system), middle (as seen in the parameter estimation domain), or interspersed throughout the learning process (as seen in the lunar lander domain).

## 4.2 Learning Performance

We now analyze the effect of INQUIRE's interaction type selections on its learning performance and compare to two types of baselines. The first, DemPref [9], learns from 3 demonstrations and then learns from preference queries by using a volume removal objective function. As our second baseline, we compare INQUIRE against agents that use only one form of interaction: demonstrations, preferences, corrections, or binary reward. Note that the preference-only agent is formulated according to [3] and thus represents this baseline method.

We first consider the changing-state formulation of each domain, where the robot is presented with a new state for each query. Since the Parameter Estimation domain does not contain states, we exclude it from this first set of results. Figure 2 illustrates this learning performance in the Linear Dynamical System and Lunar Lander domains according to three key metrics. **Distance** measures the angular distance between the ground truth feature weights ($\omega^*$) and the algorithm's estimated feature weight $\tilde{\omega}$ after each query. **Performance** measures the task reward achieved using a trajectory optimized

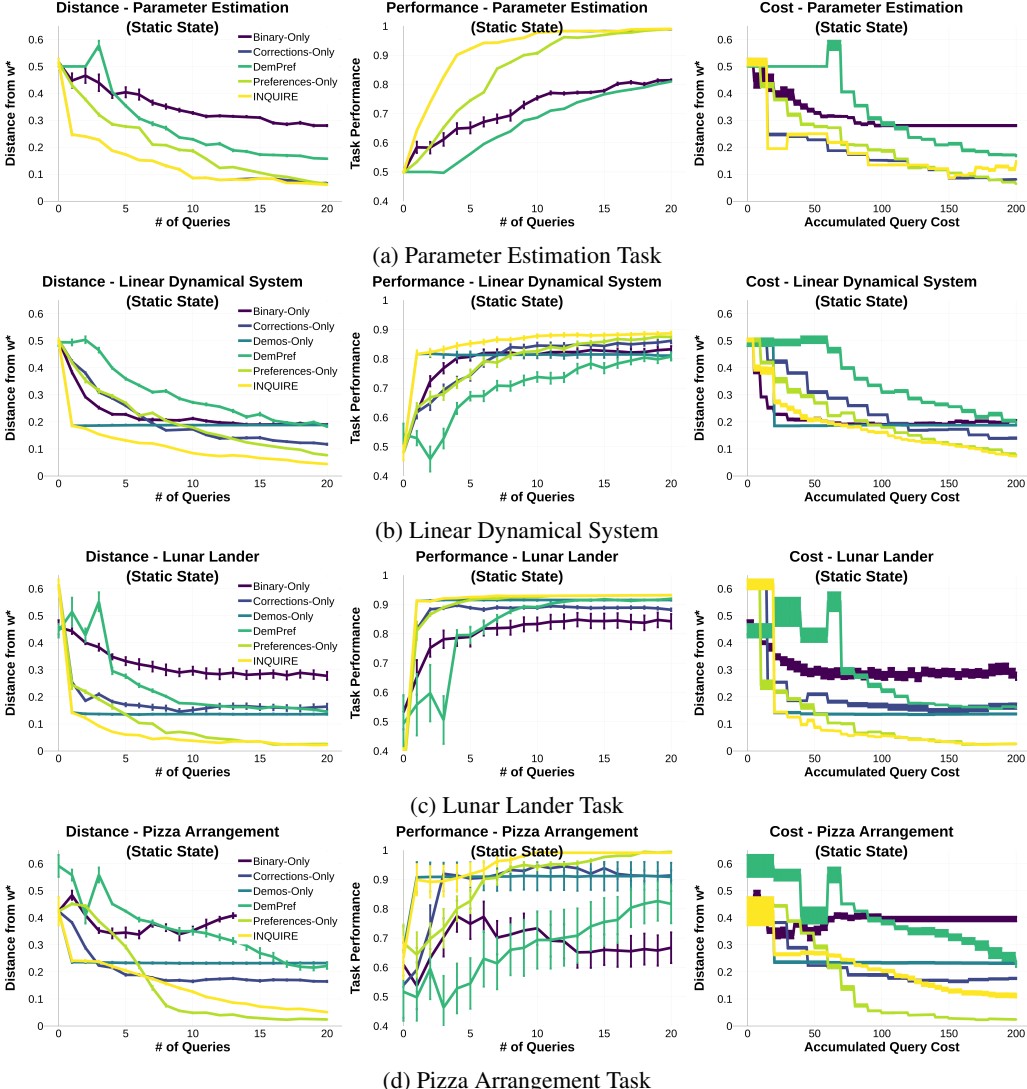

Figure 3: Metrics for the *static state* condition in which the robot is presented with the same state for all 20 queries. Error bars/regions represent variance across multiple evaluation runs with randomized query states and initial weights. Cost metrics are cut off after 20 queries for the *binary-only* method in (a) and (d) due to extensive computation times.

according to $\tilde{\omega}$ (the algorithm's estimated feature weight after each query). Performance is scaled between 0-1, with 0 and 1 representing the worst and best possible task rewards according to $\omega^*$, respectively. Note that INQUIRE's distance and performance metrics are achieved in the unweighted condition. **Cost-vs-Distance** measures the relationship between the cumulative cost of each query and the resulting distance between $\tilde{\omega}$ and $\omega^*$ after each query. INQUIRE's metrics in this graph are achieved in the cost-weighted condition.

Figure 3 presents the same three metrics for the *static-state* condition in which all 20 queries must be selected from the same initial state. Finally, we quantify these graphs by reporting the area-under-the-curve (AUC) metrics for the distance, performance, and cost curves across all tasks. These metrics are available in Appendix C. The AUC metrics indicate that, compared to the baseline methods, INQUIRE results in the best average learning performance (measured both by the distance and performance plots in Figures 2-3) across all domains and dominates learning performance in the static-state domains. INQUIRE also results in the best average distance-to-cost ratio across all domains.

## 5 Discussion

The results show the importance of dynamically selecting interaction types according to the robot's current state. For example, demonstrations can be highly informative when provided in novel states, but when the robot may only query a teacher from a single state, multiple demonstrations are likely to be very similar (if not identical). As a result, receiving multiple demonstrations in a static query state is uninformative. We see the benefits of dynamically selecting interaction types in Figure 3, where INQUIRE outperforms all single-interaction methods by optimizing both query type and content to maximize the informativeness of the query feedback.

INQUIRE selects the interaction type that, after receiving feedback, minimizes the entropy over its distribution of weight estimates $\mathcal{W}$. This distribution thus serves as a representation of the robot's current model of the task reward. Figure 1a illustrates how INQUIRE changes the query type as it learns over time (represented by # of queries). This is particularly evident in the Parameter Estimation task, where the algorithm originally requests corrections until it has refined its model of the task reward to a point where preferences become more informative (after 14-18 queries). Overall, dynamically adapting to the robot's model of the task reward results in better performance than adopting a fixed strategy for selecting interaction types (i.e., DemPref, which always requests demonstrations before selecting preferences).

An added benefit of INQUIRE is that it can incorporate a cost metric to identify cost-aware, informative queries. The AUC metrics for the cost graphs indicates that INQUIRE selects queries that, on average, minimize the cost-to-distance ratio across all domains. We expect that this cost metric is domain-specific, and can represent a number of human factors that the algorithm should take into account (e.g., the effort involved for a human to respond to each query type [8]). The cost metric used in our study thus serves as an example of how INQUIRE can factor in interaction costs.

## 6 Limitations

Our evaluation is performed using feedback from an optimal oracle. Real human feedback, however, is likely to be at least somewhat sub-optimal, and its severity likely depends on the interaction type. For example, a non-optimal demonstration may be one that is sufficient but not ideal for completing the task. In contrast, binary rewards offer only two feedback choices to the user, and thus a non-optimal binary reward may indicate the opposite information from what the user intended to convey. These examples illustrate how non-optimal feedback may need to be handled differently depending on the interaction type, and thus, should affect INQUIRE's estimation of information gain. Future work should investigate setting separate values of $\beta$ (see Eq. 3) for each interaction type, with the goal of reflecting interaction-specific expectations for sub-optimal feedback.

Furthermore, INQUIRE does not yet have the ability to select the state in which it queries the teacher. Prior work in Active Learning has shown that state selection can improve the informativeness of resulting demonstrations [23], and we expect that optimizing over the query state in addition to query type and content would improve the performance of INQUIRE.

## 7 Conclusion

We introduced INQUIRE, an algorithm enabling a robot to dynamically optimize its queries and interaction types according to its task knowledge and its state within the environment. We showed that using information gain to select not just optimal queries, but optimal interaction *types*, results in consistently high performance across multiple tasks and state configurations. Future work will include formal user studies to investigate our method's efficacy with people of varied skillsets and comfort with robots; incorporation of novel interaction types and other communication modalities; and alternative representations of the reward function and feature spaces. Moreover, we are excited at the possible extensions others might present by using our open-source framework[3] for evaluating and comparing active-learning agents across multiple environments and simulated teachers.

---

[3] https://github.com/HARPLab/inquire

**Acknowledgments**

The authors would like to thank the members of HARP Lab, IAM Lab, and the anonymous reviewers for their valuable feedback. This work is supported in part by the Office of Naval Research (N00014-18-1-2503).

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
