# OpenReview forum: "INQUIRE: INteractive Querying for User-aware Informative REasoning"
_robot-learning.org/CoRL/2022/Conference — CoRL 2022 Poster_

### Official Review · Reviewer_pRs8 · 2022-07-27

**Originality:** Fair
**Technical Quality:** Good
**Clarity Of Presentation:** Very Good
**Impact:** 3

**Recommendation:**

Weak Reject: I recommend rejecting the paper, but will not argue for my recommendation if the majority of other reviewers have a different opinion.

**Summary:**

A lot of works studied robots learning from various forms of human feedback, e.g., comparison, demonstrations, corrections, etc. Additionally, a lot of effort has been put into how to actively query humans for a specific feedback type. This paper tries to unify different feedback types so that the robot can optimize for both what type of feedback to acquire and what query (of that feedback type) to make.

The paper is well-written and mostly easy to follow. Multiple experiments have been conducted in simulated domains with simulated oracles. Baseline methods from the literature has been used.

**Issues:**

Please see my comments in the "Strengths and Weaknesses" section. I moved everything there due to character limits.

**Quality Of The Limitations Section:**

Additional details required

**Reviewer Expertise:**

5: The reviewer is absolutely certain that the evaluation is correct and very familiar with the relevant literature

**Robotics Focus:**

Highly relevant to robotics but no hardware experiments

**Strengths And Weaknesses:**

Strengths:
- The paper is trying to tackle a problem that has been discussed in the community for a relatively long time now.
- The framework allows more feedback types than any other paper in this domain (except [15]).

Weaknesses:
- The paper is trying to solve this famous problem using the most obvious approach: let the system designer decide a cost value for each feedback type. However, the difficulty of the problem is already because it is difficult to come up with those cost values. In that sense, it is not really solving the problem. If we knew cost values, this approach would already be what we would use, because it is already pointed out by earlier work (e.g., [12] or "Learning Reward Functions from Diverse Sources of Human Feedback: Optimally Integrating Demonstrations and Preferences" by Biyik et al.)
- Comparisons with baseline methods are useful and needed, but it does not really say much: the baselines are already expected to perform worse than the proposed approach because they do not have the same "action space" as the proposed approach. For example, one of the baselines is allowed to only make preference queries whereas the proposed approach can ask for demonstrations, corrections, and binary feedback in addition to preferences.
- Equations 12 and 13 are confusing. Worse than that, I am concerned the paper makes a mathematical mistake while obtaining and using the set \Omega. Please see my detailed comments below.




Below is my detailed review of the paper.


A lot of works studied robots learning from various forms of human feedback, e.g., comparison, demonstrations, corrections, etc. Additionally, a lot of effort has been put into how to actively query humans for a specific feedback type. This paper tries to unify different feedback types so that the robot can optimize for both what type of feedback to acquire and what query (of that feedback type) to make. I am very familiar with the literature in this domain, and I know this has been an open problem for some time, which many researchers have been discussing. So I appreciate the authors' effort for trying to address this problem and the contribution.

However, the paper is trying to tackle this problem through the most obvious / straightforward way: "Let the system designer decide the cost for each particular feedback type, and then maximize informativeness under some cost constraint, or maximize the difference". I think the difficulty of the problem is not in optimizing over different feedback types or learning from various forms of feedback. The difficulty is in coming up with the cost values. Hence, this paper is not really solving the problem. This is my most major concern about the paper.

Below are my other comments and criticisms:
1- This is just formatting, but is the paper scanned from hard-copy papers? For some reason, I am not able to select texts in the PDF and some portions of the PDF have lower quality. This makes reviewing on a computer a little painful (and I don't want to print the paper for obvious reasons). Please upload the LaTeX output next time.
2- In line 22 and in various other places, the paper uses the term "binary feedback" to refer to a feedback type where a human says "successful" or "unsuccessful" for a trajectory. However, "binary feedback" is a more general term than that. For example comparisons (or "preferences" as used by the paper) are also binary. I suggest using another term instead of "binary feedback". Similarly, preferences do not have to be binary: they can be, for example, rankings. But I know the term preference is usually used for pairwise comparisons, so this is okay.
3- The second paragraph of the Introduction does not mention the mental or physical effort required from the teacher while talking about the differences between the feedback types. But this is really one of the main differences here. This is later discussed, but it should be mentioned here, too, because this is the first place where differences are discussed.
4- The third paragraph says previous works typically do not work with multiple feedback types. However, [12] and the paper by Biyik et al. that I mentioned earlier handle preferences and demonstrations together.
5- I do not understand what "adapting to a state" means in the last sentence of Introduction.
6- Section 2 mentions volume removal and information gain based active querying techniques, which have really been the two most dominant techniques (even though recent works, e.g., [1], showed info gain is strictly better). But it would be nice to mention other alternatives too, e.g., max regret optimization. See "Active preference learning using maximum regret" by Wilde et al.
7- Relatedly, in general, the paper is missing many important citations, e.g.,:
7a) "Extrapolating Beyond Suboptimal Demonstrations via Inverse Reinforcement Learning from Observations" by Brown et al., which also leverages both demonstrations and rankings.
7b) "ROIAL: Region of Interest Active Learning for Characterizing Exoskeleton Gait Preference Landscapes" by Li et al., which combines preferences with ordinal labels.
7c) "APReL: A Library for Active Preference-based Reward Learning Algorithms" by Biyik et al., which introduces a software library that combines multiple different preference feedback types and demonstrations.
7d) "Unified learning from demonstrations, corrections, and preferences during physical human-robot interaction" by Mehta and Losey. The name should say it all.
8- Line 88-98 introduce an implicit assumption that the initial state is fixed during a querying iteration, so the robot cannot choose which state the demonstration should start at. This is okay, but this should be clearly and explicitly stated.
9- If the initial state is fixed, then there is no active querying in demonstrations and corrections. Is this true? If yes, then "Learning Reward Functions from Diverse Sources of Human Feedback: Optimally Integrating Demonstrations and Preferences" by Biyik et al. already proved that active feedback should be collected after non-active feedback. Why is this knowledge not utilized in selecting feedback types?
10- It seems demonstrations and corrections are mathematically equivalent, as they are non-active. Is this correct?
11- In Corr. row of Table 1 and in line 94, I believe "$q$" must be replaced with "$t$".
12- Equation 3 reduces to Bayesian IRL, so that paper by Ramachandran and Amir should also be cited.
13- The paper says $\beta$ was set to 20. This is meaningful only if there is a magnitude constraint on $\omega$. Otherwise, one can scale \omega arbitrarily and $\beta$ becomes meaningless. Please clarify any constraints on the magnitude of $\omega$.
14- The paper says $E$ is a matrix, but it is not. It is a tensor. Relatedly, what does the transpose of a tensor mean in Equation 6? Is it being transposed over the first two dimensions? Please clarify.
15- Lines 115 through 126 (and equations 4-9) are just an implementation trick, not a novelty. Although I appreciate the fact that it is shared, I think it best fits to the Appendix.
16- To calculate information gain, the framework uses a discrete set of trajectories of size $T$. For demonstrations (and corrections), this means as $T$ increases, the information gain value (which is approximated) will increase (as can be seen from Equation 9). So the information gain estimates are not unbiased, they are under-estimated. This should be discussed.
17- In Equation 10, what's the motivation for using $1/\log(\lambda_i)$ instead of a simple additive or multiplicative term? Is this somehow more interpretable or intuitive? Please elaborate.
18- Lines 136 and 137 are concerning. Do they mean the framework does not sample the set $\Omega$ from the posterior over $\omega$? The way the paper currently reads tells me that there is a bunch of samples, and then gradient ascent is applied to each sample until they converge (before the next query). What if they all go to the same local maximum (which is unlikely not to happen)? However, if the set $\Omega$ is obtained in this way, then the samples in it are not really samples from the posterior. In turn, this makes the set useless for information gain computation, because its derivation is based on the fact that $\Omega$ are the samples from the posterior.
19- The paper should cite OpenAI Gym's paper for LunarLander. I am not familiar with the Pizza Arrangement domain, but if it is from a literature, that should also be cited.
20- Experiments do not really say much, because of multiple reasons: (1) the baselines are already expected to perform worse than the proposed approach because they do not have the same "action space" as the proposed approach. For example, one of the baselines is allowed to only make preference queries whereas the proposed approach can ask for demonstrations, corrections, and binary feedback in addition to preferences. (2) The simulated humans are oracles. (3) The system has not been deployed on a real robotic system.

**Summary Of Recommendation:**

The problem is interesting, relevant and important. But the proposed framework is not addressing the main difficulty of the problem, it is introducing an obvious framework for combining different feedback types. Besides, the literature review in the paper is very limited. Finally, experiments are not realistic (e.g., oracle humans and simulated systems). All other concerns are relatively minor.

---

> ### Author Response · Authors · 2022-08-24
> **Clarifying intended contributions**
>
> Thank you for your detailed review and useful suggestions for clarifying our paper. For ease of discussion, we address each major point as a separate comment.
>
> Regarding contributions:
>
> We first want to reiterate the research problems that we do (and do not) address in this paper.
>
> We agree that determining cost values is a difficult problem. More than that, however, we believe it is an impossible problem to solve for the general case. Even determining the features relevant to such a cost function is still an unsolved problem [1], as it depends on interpersonal, safety, and physical factors of the robot's application domain. Addressing this problem requires thorough study of how various interaction types can be implemented for various robot domains, and how these implementations affect the aforementioned human factors in order to determine their quantitative and qualitative "cost". We believe that this is a very important problem, but entirely separate from the ones that we solve: (1) reasoning over the information afforded by each interaction type, (2) formulating informative queries, and (3) incorporating feedback from heterogenous interaction types into a single reward model.
>
> An important benefit of our system, in comparison to heuristic-based methods for selecting interaction types, is that we calculate the raw information gain estimates for each interaction type, and thus can weight those estimates against a domain-specific cost measure. While prior work has incorporated multiple interaction types, they do not directly estimate information gain across each type, and thus do not provide the transparency necessary to weight these estimates against a cost function. Performing this direct comparison requires (1) a formalization of the similarities and differences between interaction types as they pertain to information gain (which we contribute in terms of query space, choice space, and choice implications), and (2) a derivation of an information gain metric that incorporates these formalizations, and thus is generalizable across interaction types. As the first to perform this direct comparison using these two contributions, we contend that our work is not obvious but instead a meaningful step that is useful to the community.
>
> Overall, we do not aim to solve the problem of determining/learning cost functions. Instead, our work is the first to directly estimate information gain across multiple interaction types and, as a result, is also the first to weight informativeness against domain-specific cost metrics when choosing interaction types.
>
> [1] Y. Cui, P. Koppol, H. Admoni, S. Niekum, R. Simmons, A. Steinfeld, and T. Fitzgerald. Understanding the relationship between interactions and outcomes in human-in-the-loop machine learning. International Joint Conference on Artificial Intelligence (IJCAI), 2021.

---

> ### Author Response · Authors · 2022-08-24
> **Regarding the appropriateness of baselines**
>
> Regarding the appropriateness of baselines:
>
> Our reasons for comparing INQUIRE to single-interaction approaches are two-fold. First, it highlights the value of using a mixed-interaction strategy. Our results show that query types that are ordinarily uninformative (such as binary feedback, as seen in Figs 2-3) when used alone can be informative in conjunction with other interaction types.
>
> Second, these baselines are well-established methods for selecting informative queries for their respective interaction types. While INQUIRE has access to a larger "action space" of potential queries, it also has access to a significantly larger number of sub-optimal queries than the baseline methods. In other words, this larger “action space” only provides a benefit if the interaction selection method can accurately assess and identify the informativeness of queries, despite differences in their query space, choices available to the teacher, and implications of the teacher's feedback. Existing approaches are unable to make this assessment. Even if we were to reduce this query space to a smaller one containing only the best, most-informative queries for each interaction type, comparing these queries to each other is a (previously-unsolved) challenge. This is the challenge we have successfully met through INQUIRE.

---

> ### Author Response · Authors · 2022-08-24
> **Other clarifications**
>
> *Other Clarifications:*
>
> Regarding the uniform re-initialization of weights (comment #18):
>
> Please see our response to the meta-reviewer.
>
> 4, 6, 7, 12, 19 - We will add the requested citations. Note that the suggested paper by Mehta and Losey is not peer-reviewed and was unavailable on arxiv until after the CoRL deadline.
>
> 9 - Correct, the initial state is fixed. Corrections are actively queried, however, as the robot must propose a trajectory for the teacher to correct. We intentionally avoid the use of heuristics (such as collecting non-active feedback first) in order to adapt to the robot's actual information needs, its current state, and the domain-specific costs of each interaction type.
>
> 10 - Demonstrations are entirely passive, while a correction involves INQUIRE proposing a trajectory for the teacher to then correct. In terms of information, a demonstration can be thought of as a preference for one trajectory (the demonstrated one) over all other trajectories in T (representing trajectories that were not demonstrated). A correction can be thought of as a preference for one trajectory (the teacher’s correction of INQUIRE’s proposed trajectory) over the proposed trajectory.
>
> 13 - Beta is a value between (0, inf). While the use of this parameter (representing the teacher’s suboptimality) is common among other work using Bayesian IRL, there are no community standards for setting this value. In future work, we intend to explore the effect of this parameter on INQUIRE's interaction type selections, and whether this value should be set separately for each interaction type.
>
> 16 - As |T| increases, the probability of any particular choice in response to a demonstration or correction query (that is, P_q,c,w for some choice c) decreases. Based on Eq. 9, this results in the expected information gain of any one choice also decreasing as |T| increases, in the context of a demonstration or correction query. This effect does not occur for preference or binary reward queries since |C(q)| remains constant regardless of |T|. However, this difference is accounted for when selecting the optimal interaction type by summing over the expected information gain of all possible choices (Eq. 10).
>
> 17 - We found that converting interaction cost to a logarithmic scale resulted in a more accurate estimation of the performance-cost tradeoff across interaction types.
>
> 18 - We use two convergence thresholds: one for sampling Omega, and another for the final estimation of w. While the latter threshold is small (set to 10^-6), the former is larger (set to 10^-3) to represent a distribution rather than a local maximum. Please see our response #2 to the meta reviewer regarding the use of uniform initial sampling for weight updates.
>
> 20 - Please see our response to the meta reviewer regarding the use of oracles as teachers.
>
> *Minor Edits:*
>
> 1 - We only uploaded the LaTeX output. Some PDF readers on Ubuntu can have compatibility issues -- perhaps this is the culprit?
>
> 2 - We will replace the term “binary feedback” with “binary rewards”.
>
> 3 - We will edit the introduction to mention mental and physical effort involved in different interaction types.
>
> 5 - We will clarify this as: "INQUIRE accommodates states"
>
> 8 - We will add an earlier statement regarding the assumption that the initial state is fixed.
>
> 11 - We use “t” to reference any trajectory, whereas “q” refers to a query. In the context of line 94, this query consists of a trajectory (so they are technically interchangeable) but we reference it as “q” for consistency.
>
> 14 - Correct, it is a tensor and is transposed over the first two dimensions. We will note this in the revision.

---

> > ### Comment · Reviewer_pRs8 · 2022-08-28
> > **Additional Discussion / Clarifications**
> >
> > I thank the authors for the responses and the updates in the paper, and apologize for my late response. Overall, my thoughts regarding the contributions of the paper and some of the technical issues remain unchanged, so I will keep my score as "Weak Reject". Below, I list my responses and clarifications to some of the author responses so that authors may update their paper (for CoRL if accepted, or for another conference otherwise).
> >
> > 7- I thank the authors for expanding the discussion of the related work, and apologize for not realizing the paper by Mehta and Losey being posted after CoRL's submission deadline.
> >
> > 9- Collecting non-active feedback first is not really a heuristic, given that it is proven to be no worse than (and in practice, to be better) any other order of feedback.
> >
> > 9, 10- I thank the authors for clarification about how corrections are modeled. Previously, I did not realize corrections are treated as preferences towards the corrected trajectory over the queried trajectory.
> >
> > 13- Perhaps my comment here was not clear. I was not criticizing the use of a fixed $\beta$. I know it is pretty standard (though there has been some works that learn $\beta$ and the reward function jointly from comparisons). But using a fixed $\beta$ is meaningful only if there is a constraint on $||\omega||$. Otherwise, the learning framework can simply learn, for example, $2\omega$ instead of $\omega$ which is effectively equivalent to learning $\omega$ with $\beta=40$. So I was asking about whether there exists a constraint on $||\omega||$ -- there should be.
> >
> > 16- I disagree. I do not know if there is a misunderstanding, but let me try to clarify. A die roll carries more information than a coin toss. It is true that each possible outcome of a die roll is less likely than each possible outcome of a coin toss. But if you compute the entropies: a die roll has $\log_2(6)$ bits of information, whereas it is only $1$ bit for a coin toss. Following the similar logic in this paper's case, as $|T|$ increases, number of possible outcomes increase, so the expected information gain (after summing over all possible choices of course) increases. Therefore, using a finite $|T|$ is necessarily an underestimation of the information gain. As the authors acknowledged, this is not the case for preferences and binary feedback.
> >
> > 18- The authors' response to the meta-reviewer says they chose this way so that the order of feedback will not bias the samples. However, sampling directly from the posterior is possible, i.e., without using the samples from the prior. So in that way, there is no bias due to the order of feedback. Previous works have already done this: in fact, I am not aware of any works that use information gain and not directly sample from the posterior. Hence, although I appreciate the additional effort the authors put, it is not satisfactory, because the standard method of directly sampling from the posterior is not compared to the method used in the paper. (And I again apologize for my late response, which unfortunately does not let the authors perform further experiments for this rebuttal)

---

### Official Review · Reviewer_pgRm · 2022-07-27

**Originality:** Very Good
**Technical Quality:** Excellent
**Clarity Of Presentation:** Excellent
**Impact:** 4

**Recommendation:**

Strong Accept: I recommend accepting the paper and will argue for my recommendation even if other reviewers hold a different opinion.

**Summary:**

The paper presents a method for optimizing the information gained from interaction with a human teacher over several possible types of interaction that have been studied in prior works. The key contribution is an algorithm that uses greedy selection to determine the interaction type. This step is incorporated into an algorithm that builds on previous work in interactive robot learning to update the learner’s estimation of feature weights. An empirical evaluation of the algorithm shows how interaction types vary over the learning process, in different types of problem, and with non-uniform weighting on interaction types. Finally the manuscript demonstrates that  utilizing multi-type interactions generally outperforms a single interaction type methods.

**Issues:**

No major or minor issues other than the weaknesses outline above.

**Quality Of The Limitations Section:**

Limitations are addressed clearly

**Reviewer Expertise:**

5: The reviewer is absolutely certain that the evaluation is correct and very familiar with the relevant literature

**Robotics Focus:**

Highly relevant to robotics but no hardware experiments

**Strengths And Weaknesses:**

#### Strengths
- The paper integrates several well-founded interaction methods into a uniform framework.
- The empirical results of the interaction selection (figure 1) and its accompanying discussion are clear and support the central idea that combining interactions should lead to overall better performance by the learner.
- In comparisons to learners using single types of interactions, the paper reports multiple performance measures (alignment with ground truth weight vectors and task reward).
- Result show the algorithm presented generally outperforms comparison points in 3 of 4 tasks and has best mean learning performance of the 6 frameworks evaluated.

#### Weaknesses
- As discussed in the limitations section, an example of the algorithm implemented with a human user would be valuable. The paper does not discuss how the different interaction type requests would be communicated to the user. As this paper does seem to take this step closer to implementation ‘in the wild’, it would be helpful to address this.
- The comparison experiments utilize at least 1 demonstration to initiate the learning algorithm. However, there is little discussion of what assumptions are made about the demonstrations. Prior work has shown that suboptimal (A. Colome and C. Torras, “Dual reps: A generalization of relative entropy policy search exploiting bad experiences,” Transactions on Robotics, vol. 33, no. 4, pp. 978–985, 2017.), failed demonstrations (K. Shiarlis, J. Messias, and S. Whiteson, “Inverse reinforcement learning from failure,” 2016; D. H. Grollman and A. G. Billard, “Robot learning from failed demonstrations,” Int. Journal on Social Robotics, vol. 4, no. 4, pp. 331–342, 2012.), and even intentionally bad demonstrations (A. Kalinowska, A. Prabhakar, K. Fitzsimons, and T.D. Murphey, “Ergodic imitation: Learning from what to do and what not to do,” ICRA, pp. 3648–3654, 2021.). Would the performance of the algorithm still hold if the demonstrations were suboptimal or even negative demonstrations?


**Summary Of Recommendation:**

The paper presents an integration of several interactive learning methods. The framework could easily be built upon by others in the robotics community to incorporate other interaction modalities and improve learning through HRI.

---

> ### Author Response · Authors · 2022-08-24
> **Communicating interaction requests to the user**
>
> Thank you for your encouraging and strongly positive review. We believe that this work is timely given other recent developments in learning from various interaction types, and hope that our INQUIRE algorithm and publicly-available software implementation will help push this field forward.
>
>
> > "How would different interaction requests be communicated to the user?"
>
> This is largely domain-dependent. In all of the domains we explored, we envision the teacher receiving interaction requests from—and providing feedback through—a GUI. As an example in the Lunar Lander domain:
>
> * Demonstrations: a teacher may use the arrow keys on their keyboard to indicate a trajectory.
> * Corrections: the teacher may observe a Lunar Lander trajectory taking place and intervene when necessary to adjust it (again, using the arrow keys).
> * Preferences: the teacher would be shown two trajectories and then selects the icon indicating their favorite.
> * Binary reward: the teacher would select a thumbs up or thumbs down icon.
>
> On a physical robot, these interactions would follow those used in prior work for kinesthetic demonstrations [1] and corrections [2,3,4], while potentially using a tablet to visualize trajectories for preferences and binary feedback.
>
>
> [1] Akgun, B., Cakmak, M., Yoo, J. W., & Thomaz, A. L. (2012, March). Trajectories and keyframes for kinesthetic teaching: A human-robot interaction perspective. In Proceedings of the seventh annual ACM/IEEE international conference on Human-Robot Interaction (pp. 391-398).
>
> [2] Argall, B. D., Sauser, E. L., & Billard, A. G. (2010, August). Tactile guidance for policy refinement and reuse. In 2010 IEEE 9th International Conference on Development and Learning (pp. 7-12). IEEE.
>
> [3] Bajcsy, A., Losey, D. P., O’Malley, M. K., & Dragan, A. D. (2017, October). Learning robot objectives from physical human interaction. In Conference on Robot Learning (pp. 217-226). PMLR.
>
> [4] T. Fitzgerald, E. Short, A. Goel, and A. Thomaz. Human-guided trajectory adaptation for tool transfer. In Intl. Conf. on Autonomous Agents and MultiAgent Systems, pages 1350–1358, 2019

---

> ### Author Response · Authors · 2022-08-24
> **Regarding assumptions over demonstrations**
>
> Similar to related work, we assume that demonstrations are noisily optimal, where the amount of expected "noise" is dictated by a parameter (represented in our work as beta).
>
> In our evaluation, all interaction types are assigned the same value for beta. We are interested in performing additional analysis on this parameter in the future, namely to determine whether different interaction types should be assumed to be more or less noisy than others.
>
> We expect our algorithm to accommodate suboptimal demonstrations due to the “noisily optimal” expectation. For negative demonstrations, our algorithm does not currently enable this, but we expect it would be simple to consider these to be a separate form of interaction, such that the failed demonstration is recorded as being worse than comparable trajectories (essentially, used to update weights similar to a -1 response in the binary feedback interaction type).

---

### Official Review · Reviewer_DFDJ · 2022-07-29

**Originality:** Good
**Technical Quality:** Good
**Clarity Of Presentation:** Good
**Impact:** 3

**Recommendation:**

Weak Reject: I recommend rejecting the paper, but will not argue for my recommendation if the majority of other reviewers have a different opinion.

**Summary:**

The paper tackles interactive querying across multiple human interaction modalities for reward learning. The INQUIRE approach selects queries to maximize a cost-weighted information gain about reward feature weights across demonstration, preference, correction, and binary feedback modalities. The paper compares their approach with baselines that use a single human interaction modality or a fixed pattern of interactions in simulated environments with an optimal oracle as a human proxy. The INQUIRE method is able to often outperform baseline methods, especially when the initial state is held static.


**Issues:**

- Statistical significance of the AUC metric against baselines should be clearly analyzed.

- It is unclear how corrections differ from demonstrations in terms of information—for both settings, the choice space is the full trajectory space regardless of query. So for the queries to have different value, the human response model would need to be different. Meanwhile, looking at Eqs. 7 and 55, the distribution p(c|w) does not seem to sum to 1, making it hard to understand what the intended response model is. Further, the response model probability p(c|w) for corrections in Eq. 7 doesn’t appear to depend on the similarity between the corrected trajectory and the queried trajectory, which is inconsistent with how the oracle selects trajectories (by maximizing a ratio between the value of a trajectory and it’s dissimilarity with the query). This response model seems very inconsistent with how trajectories would be selected by a human (a corrected trajectory one state away from the query is much more likely to be selected than a completely unrelated trajectory as a correction). The authors should clarify the human response model for computing information gain for corrections, and how this model causes it to differ for the information gain.

- The binary response model seems to assign a very low probability to c=1 when there are multiple high-value trajectories in T. This is inconsistent with the oracle model, which responds with c=1 when a trajectory is in the top 75%, and could artificially deflate information gains.

- Why is gradient ascent used to update the posterior Omega? Gradient ascent could bias the posterior to place undue weight on its modes. Past works approximating information gain / volume removal with posterior sampling have used Markov chain methods to approximate posterior sampling [1].

- The authors should clarify if demonstrations and corrections need to be drawn from the randomly sampled pool T. A user making sufficiently fine corrections / demonstrations would usually want to construct a trajectory that doesn’t randomly lie within the sampled pool.

- Why is the simulated oracle deterministic rather than Boltzmann-rational like the response model for computing information gain?


[1] https://escholarship.org/content/qt88k894w7/qt88k894w7.pdf


**Quality Of The Limitations Section:**

Limitations are addressed clearly

**Reviewer Expertise:**

4: The reviewer is confident but not absolutely certain that the evaluation is correct

**Robotics Focus:**

Highly relevant to robotics but no hardware experiments

**Strengths And Weaknesses:**

Strengths:
- The writing and figures are clear and concise.
- The paper presents a clear description with matrix formulas of how the information gain across querying modalities is efficiently computed.
- The paper presents clear evaluation metrics and visualizations (# of queries and cost of queries against task performance and feature learning error; querying modality by step).
- The approach appears competitive with baselines in simulated environments.

Weakness:
- A weakness is the lack of real robot experiments and a lack of real human user studies—real humans can be significantly different from a Boltzmann-rational model and exhibit multimodal reward preferences. Further, the INQUIRE method performs best relative to baselines in - the more unrealistic static initial state settings.
- It is unclear if INQUIRE is significantly better than baseline approaches.
- It is assumed that “good” trajectories can be randomly sampled from the environment.
- The corrections and binary human response models may be unrealistic (see below).
- See other issues below.


**Summary Of Recommendation:**

The paper tackles the interesting problem of deciding not only which queries to make within a human interaction modality but also which modality to query. The paper solves this problem by elegantly extending past information-gain techniques to make cost-weighted comparisons between modalities. However, without analysis of statistical significance of simulation AUC results, real robot experiments, or human user studies it is difficult to assess how effective the INQUIRE approach is. Further, inconsistencies in the human response model for corrections and binary responses call into question the information gain computation used for these interaction modalities. The work could be significantly improved by (1) clarifying the response model for computing correction and binary response information gain, (2) conducting statistical analysis of the benefits of the approach, and (3) running experiments on real robots and/or with real human teachers.

---

> ### Author Response · Authors · 2022-08-24
> **Overview and regarding lack of user studies**
>
> Thank you for positive comments regarding the value of this work, and for the detailed suggestions to improve our paper. For ease of discussion, we address each major point in a separate comment.
>
> Regarding lack of user studies:
>
> Since this is a common topic brought up by multiple reviewers, please see our response to the meta reviewer.

---

> ### Author Response · Authors · 2022-08-24
> **Regarding unrealistic static initial state**
>
> Regarding unrealistic static initial state:
>
> We believe this “static state” is realistic for a robot that encounters the same task multiple times (yet still aims to learn/improve over time), or one that intentionally revisits a familiar task. This condition highlights the importance of assessing the best interaction type in real-time, rather than relying on heuristics about which interaction types to use first, second, and so forth.

---

> ### Author Response · Authors · 2022-08-24
> **Regarding request for statistical significance**
>
> **Comment:**
>
> Regarding statistical significance:
>
> Please see the attached figures indicating significance over the AUC results. We will add these figures to the appendix.
>
> We further want to underscore that, while we are pleased to show that INQUIRE outperforms the baselines under most conditions, this is the “cherry on top”. Our main intent is not to prove that INQUIRE always results in significantly better task performance on every task, but rather, to show generality in selecting informative interaction types across tasks. While heuristic-based approaches (e.g., only requesting demonstrations) may perform very well on some tasks (e.g., the Linear Dynamical System), they are poorly suited for other tasks (e.g., any static-state domain). Our results indicate that INQUIRE was the only method to consistently achieve high performance across all domains. Furthermore, we achieve this while also reasoning over a balance between information needs and domain-specific interaction costs, which none of the baseline methods are able to provide.
>
>
> **Zip File:**
>
> /attachment/79b1f7795f9fb258ae82ad9fa3b38b1239fa5519.zip

---

> ### Author Response · Authors · 2022-08-24
> **Regarding assumptions about trajectory sampling**
>
> Regarding assumptions about trajectory sampling:
>
> We currently use a uniform sampling approach to identify trajectory samples, similar to prior work [1,2]. In higher-dimensional problems, we expect that an RRT-style or Monte-Carlo sampling approach may be substituted instead.
>
> [1] Bobu, A., Wiggert, M., Tomlin, C., & Dragan, A. D. (2021, March). Feature expansive reward learning: Rethinking human input. In Proceedings of the 2021 ACM/IEEE International Conference on Human-Robot Interaction (pp. 216-224).
> [2] E. Biyik, M. Palan, N. C. Landolfi, D. P. Losey, and D. Sadigh. Asking easy questions: A user-friendly approach to active reward learning. In Conference on Robot Learning (CoRL), pages 1177–1190, 2020.

---

> ### Author Response · Authors · 2022-08-24
> **Answering requests for clarification**
>
> >"How do corrections differ from demonstrations in terms of information?"
>
> Demonstrations are entirely passive, while a correction involves INQUIRE proposing a trajectory for the teacher to then correct. In terms of information, a demonstration can be thought of as a preference for one trajectory (the demonstrated one) over all other trajectories in T (representing trajectories that were not demonstrated). A correction can be thought of as a preference for one trajectory (the teacher’s correction of INQUIRE’s proposed trajectory) over the proposed trajectory.
>
>
> > "Re: Eqs. 7 and 55, the distribution p(c|w) does not seem to sum to 1"
>
> Note that for corrections, each query consists of a single proposed trajectory t \in T. The choice space (c) consists of any other trajectory in T that the teacher may correct the robot’s motion to. The resulting probability of choosing correction c in response to query t is thus equivalent to the likelihood of a preference query (Eq 52) where the teacher has already indicated their preference.
>
>
> > "p(c|w) for corrections in Eq. 7 doesn’t appear to depend on the similarity between the corrected trajectory and the queried trajectory, which is inconsistent with how the oracle selects trajectories "
>
> Correct. We have not yet incorporated distance metrics into the probability model itself, but expect that this could further improve INQUIRE’s ability to learn from corrections.
>
>
> > "The binary response model seems to assign a very low probability to c=1 when there are multiple high-value trajectories in T. This is inconsistent with the oracle model. "
>
> Correct. In defining the oracle model, we set the threshold for allocating positive/negative rewards as a parameter (in our evaluations, that threshold is 75%). However, the robot does not know what this threshold is, and so we adopt a more conservative probability model. In future work, we are interested in evaluating the effect of this threshold on learning performance, as well as exploring how INQUIRE may be able to estimate this threshold value as it receives feedback over time.
>
>
> > "Why is gradient ascent used to update the posterior Omega?"
>
> Please see our response to the meta-reviewer.
>
>
> > "The authors should clarify if demonstrations and corrections need to be drawn from the randomly sampled pool T. "
>
> We generate two trajectory sample sets: one for INQUIRE (to estimate information gain and update weights from feedback) and one for the oracle (to select its query response). Demonstrations and corrections must be selected from the sample set, but note that the oracle’s response may lie outside the set used by INQUIRE for selecting and learning from queries.
>
>
> > "Why is the simulated oracle deterministic rather than Boltzmann-rational like the response model for computing information gain?"
>
> Boltzmann rationality makes sense in the context of one query type; for example, a teacher that only provides demonstrations may simulate sub-optimality by sampling a trajectory with probability defined by a Boltzmann-rational model. Since it is difficult to guarantee that the degree of sub-optimality would be consistent across all interaction types, modeling a single sub-optimal teacher that unifies multiple query types is a larger challenge.
>
> The core focus of this paper is to demonstrate INQUIRE’s ability to reason over information that can be gained from multiple interaction types. Therefore, to isolate the agent’s learning ability from inconsistencies in modeling a teacher’s sub-optimality, we opt to model the oracle as one that provides optimal, deterministic feedback. That said, we recognize the importance of handling sub-optimal feedback [Section 6, Limitations] and would hope to explore this further in future work.

---

### Official Review · Reviewer_rWdS · 2022-08-05

**Originality:** Very Good
**Technical Quality:** Excellent
**Clarity Of Presentation:** Excellent
**Impact:** 4

**Recommendation:**

Weak Accept: I recommend accepting the paper, but will not argue for my recommendation if the majority of other reviewers have a different opinion.

**Summary:**

This paper formulates a method for active reward learning from user feedback, where the agent’s goal is to learn the task reward function, while also having control over what type of feedback to query from the user. Four types of feedback are considered: demonstrations, preferences between two trajectories, trajectory corrections, and binary good/bad labels. The paper formulates the choice for feedback queries as an optimization to maximize information gain in the distribution over reward, which is estimated via a discrete set of weight hypotheses.
Results are presented using a simulated oracle with access to a large set of demonstrations and their rewards to provide feedback using the available modalities. The paper shows very interesting results, where INQUIRE tends to prefer demonstrations initially, and later switches to preferences and other types of feedback, thereby outperforming a learning from demonstration while using less expensive type of supervision.


**Issues:**

Questions:
- It seems a little unintuitive that in the unweighted case the agent often selects preference queries over demonstrations and corrections. Why are they considered more informative?
- If the feedback provider was able to observe the behavior of the policy in practice over time, like a human bystander would, could they provide more informative demonstrations that target the current failure modes?
- Would it make sense to compare to a baseline that emulates an online imitation learning algorithm such as DAgger? For example, by rolling in a policy that optimizes the current most likely weights, and then requesting a correction from some random starting state. It seems like one of the obvious approaches to building an online learning system in cases where trajectory-level supervision is possible.

Suggestion:
- Add some visualizations that better explain the specific tasks and give an intuitive understanding of complexity?

**Quality Of The Limitations Section:**

Limitations are addressed clearly

**Reviewer Expertise:**

3: The reviewer is fairly confident that the evaluation is correct

**Robotics Focus:**

Relevant but unlikely to deploy to hardware in near future

**Strengths And Weaknesses:**

Strengths:
- The paper looks at an interesting, relevant and under-studied problem.
- The method is very clean and well motivated.
- Paper is well written with clear problem formulation, method description, notation etc.
- Extensive experiments show effectiveness of the method on an appropriate set of simulated tasks.

Weaknesses:
- It seems that the formulation assumes a reasonably low-dimensional reward, query, and answer space. I am not sure how well this method would scale to more complex high-dimensional tasks, such as manipulation.
- Did not test on a real-robot problem.


**Summary Of Recommendation:**

The paper proposes a well-formulated method to a relevant and well-motivated problem. However, the experimental domains were somewhat simple which casts doubt on the applicability to more complex robotics tasks.

---

> ### Author Response · Authors · 2022-08-24
> **Overview and regarding scalability and the use of a low-dimensional feature space**
>
> Thank you for taking the time and effort to review our work. We appreciate your overall positive feedback, and use this opportunity to provide the clarifications you have requested. For ease of discussion, we address each major point in a separate comment.
>
> Regarding scalability and the use of a low-dimensional feature space:
> INQUIRE learns a reward function, which we expect would scale to a manipulation problem through the use of a sampling-based planner (using the reward function as an objective), or in the same fashion as [1].
>
> While we do not evaluate a truly high-dimensional problem (such as learning weights for a neural network), our evaluation tasks involve learning 4-8 feature weights, which is comparable to other papers in this research space. Please see the attached PDF for a table comparing the dimensionality of related work.
>
> [1] Bobu, A., Wiggert, M., Tomlin, C., & Dragan, A. D. (2021, March). Feature expansive reward learning: Rethinking human input. In Proceedings of the 2021 ACM/IEEE International Conference on Human-Robot Interaction (pp. 216-224).

---

> ### Author Response · Authors · 2022-08-24
> **Regarding the selection of preference queries in the unweighted case**
>
> Regarding the selection of preference queries in the unweighted case:
>
> Demonstrations and corrections enable an infinite number of possible responses by the teacher, whereas preference queries are bounded by the two choices offered by the robot. INQUIRE accounts for the expected information gain of any potential query response by the teacher, which is then weighted by the likelihood of the teacher choosing those responses. As a result, it accounts for potentially uninformative responses to a demonstration or correction (since their response spaces are unbounded), whereas the preference queries are selected by INQUIRE to maximize informativeness regardless of which preference the user selects.

---

> ### Author Response · Authors · 2022-08-24
> **Regarding teachers targeting failure modes**
>
> Regarding teachers targeting failure modes:
>
> This is an interesting point, which may relate to scaffolding processes used by human teachers (i.e. where teachers adaptively instruct a learner by first teaching one skill and then -- upon perceived mastery -- teaching another). In our setting, we do not assume or address such sophisticated teaching approaches. Instead, we make the assumption of a lay instructor who gives only positive examples of desired behavior. In future studies related to this, we would instruct participants to provide feedback in this way. However, allowing for this type of behavior could certainly be an interesting extension of this work that would enable adaptation to a wider set of teaching strategies.

---

> ### Author Response · Authors · 2022-08-24
> **Regarding DAgger as a baseline**
>
> Regarding DAgger as a baseline:
>
> An algorithm such as DAgger could provide an alternate way to implement the "corrections" interaction type, since the learning agent generates a trajectory by rolling out the current learned policy, and then presents this trajectory to a teacher to receive a modified trajectory based on an expert policy. In this regard, DAgger would become a baseline as a corrections-only agent. The baseline evaluations presented in this paper were primarily to demonstrate the effectiveness and importance of being able to reason over multiple interaction types; though exploring other ways to implement interactions (and by extension, other ways to implement single-interaction baselines) would be interesting for future work.

---

> ### Author Response · Authors · 2022-08-24
> **Regarding the request for visualizations**
>
> Regarding the request for visualizations:
>
> Please see the attached PDF containing visualizations that we will add to the paper’s appendix.

---

### Meta-Review · Area_Chair_v95V · 2022-08-13

**Recommendation:** Accept (Poster)
**Confidence:** 3

**Metareview:**

Summary:

In this paper, we propose a method for estimating which robot is the target of human attention in a situation where a multi-agent robot and a human are interacting by using the information quantity Transfer Entropy. The proposed method enables active estimation, replacing conventional methods for eye detection.

Strength:

- The paper is well written with precise problem formulation, method description, notation, etc.
- The paper presents a clear description with matrix formulas of how the information gain across querying modalities is efficiently computed.
- Results show the algorithm presented generally outperforms comparison points in 3 of 4 tasks and has the best mean learning performance of the six frameworks evaluated.

Weakness:

- A weakness is the lack of actual robot experiments and a lack of real human user studies—real humans can be significantly different from a Boltzmann-rational model and exhibit multimodal reward preferences.
- The paper does not discuss how the different interaction type requests would be communicated to the user.
- The comparison experiments utilize at least one demonstration to initiate the learning algorithm. However, there is little discussion of what assumptions are made about the demonstrations.
- It is unclear if INQUIRE is significantly better than baseline approaches.
- Some visualizations would be required to give a better explain the specific tasks and an intuitive understanding of the complexity.


**Best Paper Nomination:**

No

---

> ### Author Response · Authors · 2022-08-24
> **Regarding sub-optimal teachers and lack of user study**
>
> Thank you to all the reviewers for taking the time and effort to provide us with useful feedback. We have made clarifications and performed additional analyses that we believe strengthen our paper.
>
> A common concern brought up by the reviewers is the paper’s lack of a user study. We agree that this is an important next step for this work, and expect that the purpose of this study would be to validate our methods for estimating the information gain of feedback from real users. However, we argue that this is outside the scope of a single conference paper for two reasons.
>
> 1) Our approach for estimating information gain is based primarily on the Boltzmann-rational equation, which has been validated in the cognitive science literature [1] and through user studies [2,3,4] for modeling human intent from demonstrations and preferences.
>
> 2) We focus on reasoning over the information that *can* be derived from interaction types, but note that the actual implementation of this interaction will affect the quality of that data and thus any ML system’s ability to learn from it [5]. For example, a demonstration of a robot arm trajectory can be provided through a joystick or by physically moving the arm to a new position. Both result in the same information potential (i.e., both provide a demonstration) but the joystick demonstration will likely be noisier due to the challenge of the teacher mapping their intended, high-dimensional trajectory to a low-dimensional joystick input. In order to disentangle INQUIRE’s ability to assess information gain over interaction types from the effect of interaction implementation details, we assess only the former in this paper.
>
> [1] L. Baker, J. B. Tenenbaum, and R. R. Saxe. Goal inference as inverse planning. In Proceedings of the Cognitive Science Society, volume 29, 2007.
>
> [2] E. Biyik, M. Palan, N. C. Landolfi, D. P. Losey, and D. Sadigh. Asking easy questions: A user-friendly approach to active reward learning. In Conference on Robot Learning (CoRL), pages 1177–1190, 2020.
>
> [3] D. Ziebart, A. L. Maas, J. A. Bagnell, and A. K. Dey. Maximum entropy inverse reinforcement learning. In AAAI, volume 8, pages 1433–1438, 2008.
>
> [4] B. Ibarz, J. Leike, T. Pohlen, G. Irving, S. Legg, and D. Amodei. Reward learning from human preferences and demonstrations in atari. NeurIPS, 31:8011–8023, 2018.
>
> [5] Y. Cui, P. Koppol, H. Admoni, S. Niekum, R. Simmons, A. Steinfeld, and T. Fitzgerald. Understanding the relationship between interactions and outcomes in human-in-the-loop machine learning. International Joint Conference on Artificial Intelligence (IJCAI), 2021.

---

> ### Author Response · Authors · 2022-08-24
> **Regarding the use of re-initialized weights during learning**
>
> Section 3.2 summarizes how INQUIRE updates its weights according to all feedback received thus far. To do so, it initializes weights randomly before performing gradient ascent using a gradient derived from the total set of feedback. We perform this re-initialization, rather than using the previous weight samples $Omega$, because we expected it would prevent the order of feedback from biasing the weight samples. To address the reviewers’ concerns, we have now directly compared the performance of INQUIRE when using re-intialized weights vs retaining the weight sample learned from the previous feedback only. We present this comparison in the attached PDF. Overall, the results are similar across both variations of INQUIRE, with the exception of slightly better performance on the Pizza task when using the retained weights. We are interested in analyzing this effect further in future work, and how it may result in even better overall performance by INQUIRE.